# Evaluating algorithmic fairness of machine learning models in predicting underweight, overweight, and adiposity across socioeconomic and caste groups in India: evidence from the longitudinal ageing study in India

John Tayu Lee[1,2,3]*, Sheng Hui Hsu[2], Vincent Cheng-Sheng Li[3], Kanya Anindya[4], Meng-Huan Chen[3], Charlotte Wang[5], Toby Kai-Bo Shen[3], Valerie Tzu Ning Liu[3], Hsiao-Hui Chen[3], Rifat Atun[1]

1 Department of Global Health and Population, Harvard T.H. Chan School of Public Health, Harvard University, Boston, Massachusetts, United States of America, 2 Master Program in Statistics of National Taiwan University, Taipei, Taiwan, 3 Institute of Health Policy and Management, College of Public Health, National Taiwan University, Taipei, Taiwan, 4 School of Public Health and Community Medicine, University of Gothenburg, Gothenburg, Sweden, 5 Institute of Health Data Analytics and Statistics, College of Public Health, National Taiwan University, Taipei, Taiwan

* johntayulee@ntu.edu.tw

## Abstract

Machine learning (ML) models are increasingly applied to predict body mass index (BMI) and related outcomes, yet their fairness across socioeconomic and caste groups remains uncertain, particularly in contexts of structural inequality. Using nationally representative data from more than 55,000 adults aged 45 years and older in the Longitudinal Ageing Study in India (LASI), we evaluated the accuracy and fairness of multiple ML algorithms—including Random Forest, XGBoost, Gradient Boosting, LightGBM, Deep Neural Networks, and Deep Cross Networks—alongside logistic regression for predicting underweight, overweight, and central adiposity. Models were trained on 80% of the data and tested on 20%, with performance assessed using AUROC, accuracy, sensitivity, specificity, and precision. Fairness was evaluated through subgroup analyses across socioeconomic and caste groups and equity-based metrics such as Equalized Odds and Demographic Parity. Feature importance was examined using SHAP values, and bias-mitigation methods were implemented at pre-processing, in-processing, and post-processing stages. Tree-based models, particularly LightGBM and Gradient Boosting, achieved the highest AUROC values (0.79–0.84). Incorporating socioeconomic and health-related variables improved prediction, but fairness gaps persisted: performance declined for scheduled tribes and lower socioeconomic groups. SHAP analyses identified grip strength, gender, and residence as key drivers of prediction differences. Among mitigation strategies, Reject Option Classification and Equalized Odds Post-processing

**Data availability statement:** The data that support the findings of this study are publicly available from https://github.com/johntayulee-HEPI/LASI-BMI.git.

**Funding:** JTL was partially supported by the Mount Jade Project Yushan Fellow Program of the Ministry of Education (MOE), Taiwan (MOE-112-YSFMN-0003-002-P1). This work was also supported by the National Health Research Institutes (NHRI), Taiwan, through the Innovative Research Grant (IRG); Grant No: NHRI-EX114-11421PI. The funders had no role in the study design, data collection and analysis, decision to publish, or preparation of the manuscript.

**Competing interests:** The authors have declared that no competing interests exist.

moderately reduced subgroup disparities but sometimes decreased overall performance, whereas other approaches yielded minimal gains. ML models can effectively predict obesity and adiposity risk in India, but addressing bias is essential for equitable application. Continued refinement of fairness-aware ML methods is needed to support inclusive and effective public-health decision-making.

## Author summary

India now faces the paradox of widespread under-nutrition alongside a rising tide of obesity among its older population. We asked whether state-of-the-art machine-learning models could accurately identify individuals at highest risk of under-weight, overweight–obesity, and central adiposity while treating all social groups equitably. Using nationally representative data on more than 55,000 adults aged 45 years and above, we compared gradient-boosted decision trees, random forests, logistic regression, and other approaches with conventional regression techniques. Overall, the modern algorithms produced the strongest predictions. Yet a closer look revealed systematic shortfalls for scheduled tribes, scheduled castes, and the lowest income quintile—even when the models achieved excellent accuracy in the population as a whole. We then applied several well-established bias-mitigation strategies, such as re-weighting the training data and post-processing the decision thresholds. These interventions reduced the performance gap for disadvantaged groups, albeit at a modest cost to overall accuracy. By combining careful fairness audits with Shapley-based interpretation of feature importance, we illuminate how socioeconomic and caste-related factors shape both nutritional risk and prediction error. Our findings underscore that fair, trustworthy decision support systems in public health must be designed explicitly with equity objectives, rather than assuming that technical excellence alone will guarantee just outcomes.

## Introduction

India, the most populous country in the world, is undergoing a profound demographic and epidemiological transition, and this transformation remains uneven across regions and socioeconomic groups [1,2]. The prevalent issues of underweight, overweight, obesity, and adiposity are major contributors to these health inequalities and closed linked to country's rapid demographic and epidemiological transitions [3–5]. These conditions range from malnutrition to overnutrition and mirror shifts in lifestyle, economic development, and societal changes throughout the nation [1,6,7]. This complex nutritional landscape imposes a significant burden of premature mortality and hinders progress toward multiple Sustainable Development Goals (SDGs), including those targeting good health and well-being (SDG 3), zero hunger (SDG 2), and reduced inequalities (SDG 10), as well as other interconnected goals such as quality education (SDG 4) [8].

This study focuses on adults aged 45 years and older in line with the design of the Longitudinal Ageing Study in India (LASI) [9]. In later life, India's "double burden" of malnutrition is especially salient as age-related changes in body composition and functional status intersect with multimorbidity, and national data document the coexistence of underweight and adiposity in older adults, with high-risk waist circumference particularly common among women [10]. At the same time, National Family Health Survey (NFHS-5) results show rising overweight/obesity in younger adults, indicating distinct age patterning across the life course [11]. Together, these patterns motivate an equity-oriented focus on older adults and align with India's National Programme for Health Care of the Elderly (NPHCE) [12].

Early and accurate prediction of underweight, overweight, obesity, and adiposity is crucial for the effective implementation of public health interventions [13]. Extensive research has shown that the factors influencing these conditions vary significantly across India's diverse populations [14–16]. This variability poses a challenge for traditional epidemiological methods, which often struggle to capture the complex interactions among sociodemographic, household, environmental, and lifestyle factors in detail. Machine learning approaches offer new solution with its advanced algorithms capable of adapting to and analyzing datasets with a wide range of interconnected variables. These methods not only improve our ability to characterize and predict body mass index (BMI) [17] and anthropometric outcomes [18] but do so with exceptional precision.

Moreover, in a variety of clinical domains, we can see that machine learning (ML) have led to remarkable improvements in predicting health outcomes—from diagnosis to risk stratification. For example, Rajkomar et al. (2018) [19] argued that ML systems have the potential to advance health equity if fairness is incorporated from design through deployment. Obermeyer et al. (2019) [20] demonstrated that widely used healthcare algorithms could exhibit substantial racial bias, leading to the under-identification of high-risk patients among marginalized groups. Despite these early insights, many subsequent studies have predominantly focused on overall predictive performance rather than rigorously assessing and mitigating bias across sensitive subgroups. A systematic review by Chen et al. (2024) [21] highlighted that although numerous methods exist to detect and reduce bias in machine learning models. It has been suggested that bias in machine learning models can arise from various sources, including the training data, the design of the algorithms, and the methods used to interpret data [19,21]. Such biases could potentially exacerbate existing disparities in health outcomes across different demographic groups, including variations across castes, and socioeconomic statuses [22–24].

Accordingly, this study evaluates the fairness in machine learning algorithm in predicting these BMI and central adiposity in India adult population. Specifically, the objectives are to (1) assess the accuracy and fairness of ML models across different socioeconomic and caste groups, (2) compare ML performance with traditional regression methods, (3) examine the impact of incorporating socioeconomic and household variables on model accuracy and fairness, (4) identify key predictive features influencing BMI and central adiposity, (5) investigate the extent and source of bias in ML predictions across socioeconomic and caste groups, and (6) evaluate the effectiveness of various bias mitigation techniques in improving fairness without compromising model performance.

## Results

### Descriptive summary

The analytic sample comprised 55,647 adults aged 45–116 years (mean [SD] 59.4 [10.4]); 46.67% were men and 53.33% were women. By caste, 27.58% were General, 17.64% Scheduled Caste, 16.88% Scheduled Tribe, and 37.90% Other Backward Classes. Monthly per-capita consumption expenditure (MPCE) quintiles were approximately evenly distributed: Lowest 20.28%, Lower-middle 20.33%, Middle 19.99%, Upper-middle 19.81%, and Highest 19.59%. Overall, 18.5% (n = 10,315) were underweight, 44.2% (n = 24,570) were overweight/obese, and 45.9% (n = 25,515) had high waist circumference.

Outcome composition differed across subgroups. Underweight was older-skewed relative to the full sample (e.g., ages 65–75: 27.05% vs 22.07%; 75–85: 12.09% vs 7.59%; ≥85: 3.68% vs 1.82%), concentrated among those with no schooling (64.93%) and the lowest MPCE quintile (32.39%), and over-represented in the East (26.07%) and Central (21.80%)

regions. In contrast, overweight/obesity and high waist circumference were concentrated in ages 45–65 (e.g., 45–55: 42.55% and 39.28%; 55–65: 32.29% and 32.54%), disproportionately female (58.18% and 73.79%, respectively), more represented in higher MPCE groups (overweight/obesity 27.17% and high waist 25.53% in the highest quintile), and comparatively higher in the South (30.57% and 28.48%) and North (21.39% and 22.48%). By caste, adiposity outcomes were more represented among the General category (e.g., overweight/obesity 34.56%), whereas underweight was relatively more prevalent among SC (21.88%) and ST (20.97%). (S1 Table)

**Predictive performance of machine learning models**

The predictive performance of various machine learning models for identifying underweight, overweight/obesity, and high waist circumference among the Indian population is summarized in S3 Table. Fig 1 presents the ROC curves and corresponding AUROC values for these models. Notably, the ROC curve visualization and AUROC in the plot are based on the overall model applied to the test data without bootstrap resampling.

**Underweight prediction.** For underweight prediction, the AUROC values ranged from 0.72 to 0.80, with the highest performance observed in Logistic Regression, Gradient Boosting, and LightGBM, all achieving an AUROC of 0.80. While these models exhibited strong specificity, sensitivity remained consistently low across all groups. This discrepancy may be attributed to the imbalance in the data, as the prevalence of underweight cases was lower compared to other health outcomes. Additionally, models achieved higher accuracy for individuals with high MPCE compared to those with low MPCE; however, no notable difference was observed in AUROC.

**Overweight/Obesity prediction.** For overweight/obesity prediction, the AUROC values ranged from 0.70 to 0.80, with LightGBM and Logistic Regression outperforming other models. However, the model's performance varied across subgroups based on MPCE. In the lowest MPCE subgroup, LightGBM achieved a sensitivity of 0.37 (95% CI: 0.33–0.41) and a specificity of 0.92 (95% CI: 0.91–0.94), indicating limited ability to identify overweight/obesity cases despite strong non-case classification. Conversely, in the highest MPCE subgroup, sensitivity improved to 0.84 (95% CI: 0.82–0.86), but specificity dropped to 0.51 (95% CI: 0.47–0.54), highlighting a trade-off between case detection and non-case accuracy.

**High waist circumference prediction.** For high waist circumference prediction, AUROC ranged from 0.74 to 0.84, with LightGBM, Gradient Boosting, and Logistic Regression again performing best. Similar to overweight and obesity prediction, the performance of LightGBM varied considerably across MPCE subgroups in terms of sensitivity and specificity, while accuracy remained relatively consistent. In the lowest MPCE subgroup, the model achieved a sensitivity of 0.52 (95% CI: 0.48–0.56) and a specificity of 0.89 (95% CI: 0.88–0.91). However, in the highest MPCE subgroup, sensitivity increased to 0.83 (95% CI: 0.80–0.85), while specificity declined to 0.63 (95% CI: 0.60–0.66), reflecting a trade-off between improved case detection and reduced non-case classification accuracy.

**Model assessment across outcomes.** Across all outcomes, Logistic Regression and LightGBM were identified as the most robust model overall, while neural network models consistently demonstrated lower performance. Due to its superior predictive capability across all outcomes and its ability to capture complex nonlinear relationships, LightGBM was selected as the primary model for further analysis.

**Prediction density analysis**

Fig 2 shows the density plots of predicted probabilities for the positive (blue) and negative (orange) classes, stratified by caste and MPCE. Well-separated curves—often appearing as distinct double peaks—signal strong discrimination (high sensitivity and specificity), whereas overlapping curves indicate poorer performance.

For underweight predictions, the positive and negative curves largely overlap across all subgroups, with only a slight left skew in the positive distribution, indicating weak discrimination. This is consistent with the lower sensitivity and specificity reported in S2 Table for underweight predictions. In contrast, the overweight/obesity plots reveal more pronounced

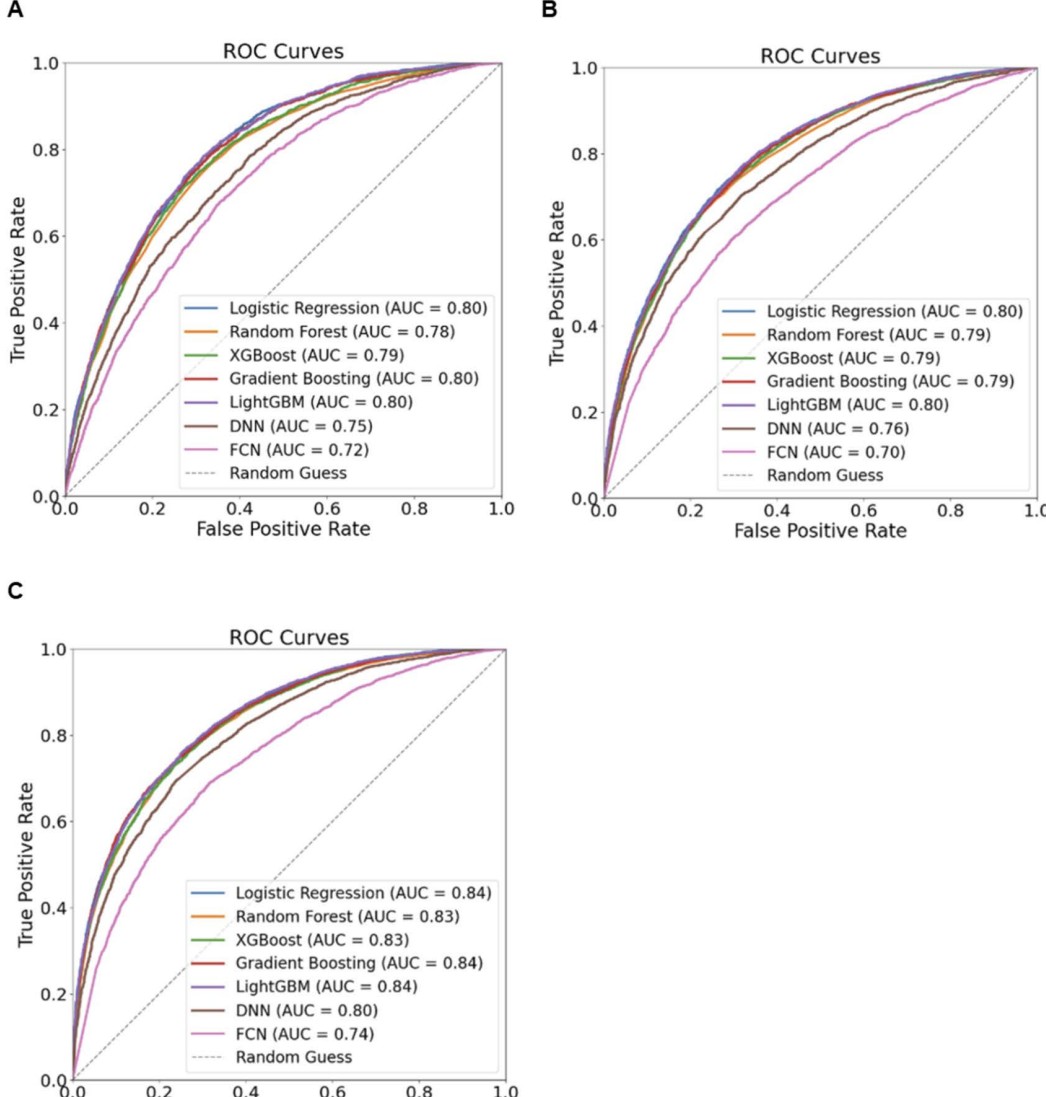

**Fig 1. Receiver Operating Characteristic (ROC) Curve with Area Under the Curve (AUROC) for Model Performance Evaluation.** (a) Underweight, (b) Overweight/Obesity, (c) High Waist Circumference.

double peaks—especially in higher MPCE subgroups and among Other Backward Classes and General caste—though Scheduled Caste, Scheduled Tribe, and the lowest MPCE subgroup exhibit more overlap. A similar trend is observed for high waist circumference, where moderate to high MPCE subgroups show distinct double peaks, while the lowest MPCE subgroup displays considerable overlap; notably, the middle MPCE subgroup presents the clearest double peaks, suggesting the best balance between sensitivity and specificity.

Taken together, these density plots confirm that LightGBM discriminates overweight/obesity and high waist circumference more effectively than underweight, although performance varies by caste and socioeconomic status, with the poorest separation observed in the lowest MPCE and marginalized caste groups.

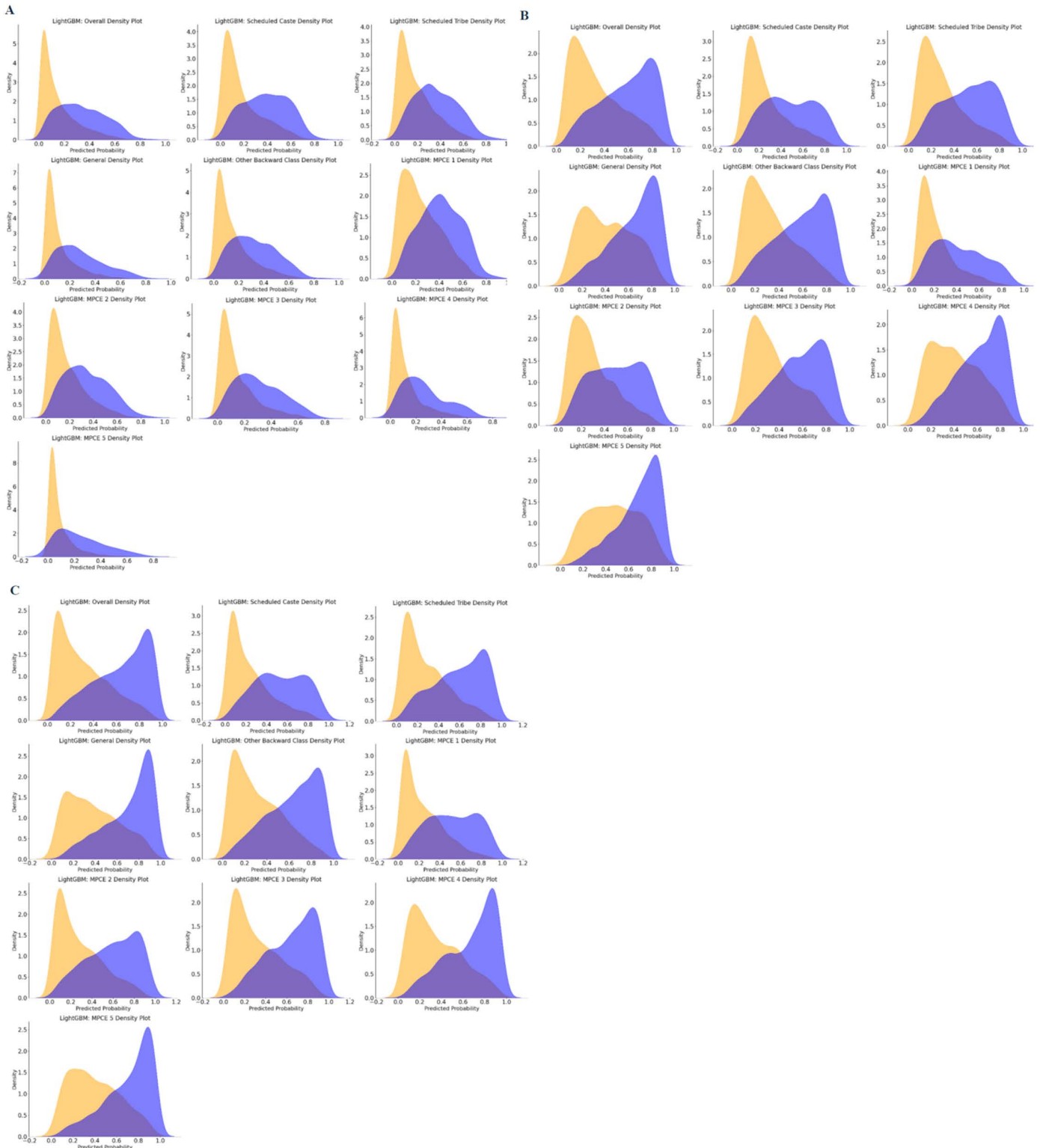

**Fig 2. Density Distribution of Model Predictions by Subgroups Stratified by Caste and Monthly Per Capita Expenditure (MPCE) Using LightGBM.**

## Incorporating socioeconomic and health features

Table 1 presents the performance metrics of the predictive models after incorporating both socioeconomic and health-related variables. For the underweight prediction model, the inclusion of these variables resulted in an increase in the AUROC from 0.74 to 0.75, and further to 0.78 after adding health-related variables. Similarly, for the overweight prediction, the AUROC increased from 0.75 to 0.76 with the addition of socioeconomic variables, and further to 0.78 after incorporating health-related variables. In the case of waist circumference, the AUROC improved from 0.80 to 0.81 with the inclusion of socioeconomic variables, and further to 0.82 with the addition of health-related data. These findings underscore the critical role of integrating socioeconomic and health-related factors to enhance the predictive performance of the models.

## Feature importance analysis

Fig 3 presents a summary plot of Shapley values for feature importance analysis. The most influential features were grip strength, gender, and residence, as indicated by the highest Shapley values. Additionally, hypertension, diabetes, and education level were also found to be significant contributors to the model predictions. These features played a crucial role in the overall performance of the predictive models.

Focusing on the top three features, we observed that grip strength had a negative impact on underweight individuals and a positive impact on overweight individuals and those with high waist circumference. Residence in rural areas was associated with a higher likelihood of underweight, while urban residents were more likely to be overweight or have a high waist circumference. Lastly, males were more likely to be underweight, while females were more likely to be overweight, with females also showing a higher likelihood of having a high waist circumference.

## Bias mitigation in socioeconomic and caste subgroup

Fig 4, S4 Table, and S5 Table compared multiple bias mitigation strategies for the three target outcomes—underweight (Fig 4A1/A2), overweight/obesity (Fig 4B1/B2), and high waist circumference (Fig 4C1/C2)—by presenting box plots for three key performance metrics (AUROC, sensitivity, and specificity) across caste and MPCE subgroups. In these plots, the grey boxes represent the original best-performing model (LightGBM), while the colored boxes correspond to various bias mitigation techniques. Narrower boxes indicate smaller variability among subgroups (i.e., less disparity), and higher median values reflect stronger overall performance. Notably, Reject Option Classification and Equalized Odds

**Table 1. Model Performance Metrics After Integrating Socioeconomic and Health-Related Variables Using LightGBM.**

|  | AUROC | Accuracy | Sensitivity | Specificity | Precision |
|---|---|---|---|---|---|
| **Underweight** | | | | | |
| Demographic | 0.7387 | 0.8100 | 0.0659 | 0.9856 | 0.5185 |
| + Socioeconomic | 0.7497 | 0.8144 | 0.0998 | 0.9830 | 0.5808 |
| + Health-related | 0.7756 | 0.8173 | 0.1482 | 0.9751 | 0.5844 |
| **Overweight/Obesity** | | | | | |
| Demographic | 0.7493 | 0.6911 | 0.6013 | 0.7619 | 0.6659 |
| + Socioeconomic | 0.7612 | 0.7010 | 0.6203 | 0.7647 | 0.6753 |
| + Health-related | 0.7817 | 0.7140 | 0.6407 | 0.7719 | 0.6891 |
| **High Waist Circumference** | | | | | |
| Demographic | 0.8050 | 0.7286 | 0.6534 | 0.7925 | 0.7284 |
| + Socioeconomic | 0.8136 | 0.7338 | 0.6720 | 0.7864 | 0.7282 |
| + Health-related | 0.8271 | 0.7487 | 0.6976 | 0.7922 | 0.7409 |

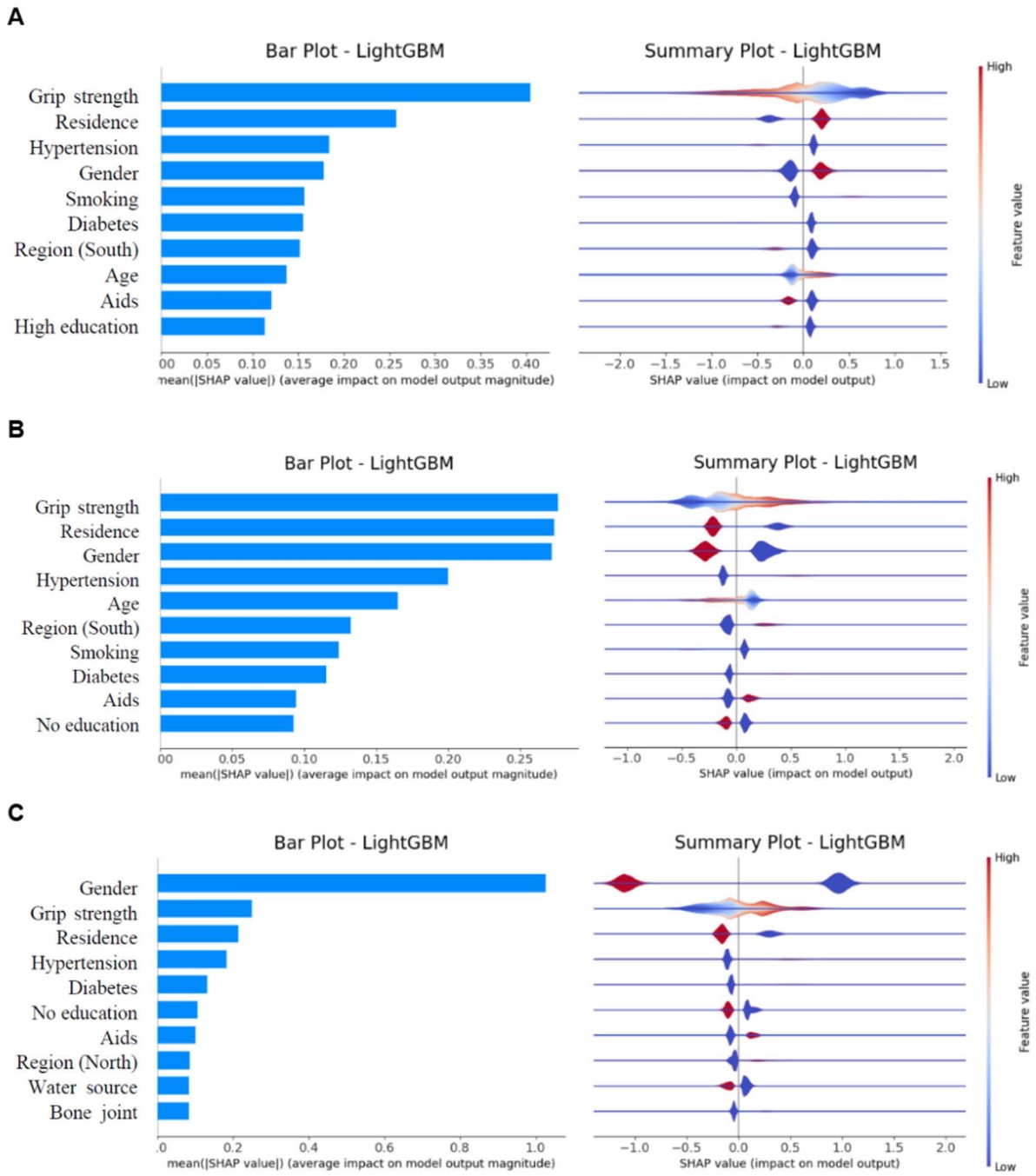

**Fig 3. Feature Importance Analysis and Summary Plot of Shapley Values for Model Interpretability.**

Postprocessing (as seen in Fig 4A1 and 4A2) tend to "pull up" the low-performing subgroups and "pull down" the high-performing ones, which narrows the boxes and results in more uniform performance across caste and MPCE; however, this uniformity is sometimes achieved at the expense of lower median values, highlighting a trade-off between fairness and predictive power. In contrast, while Exponentiated Gradient Reduction and Adversarial Debiasing also reduce

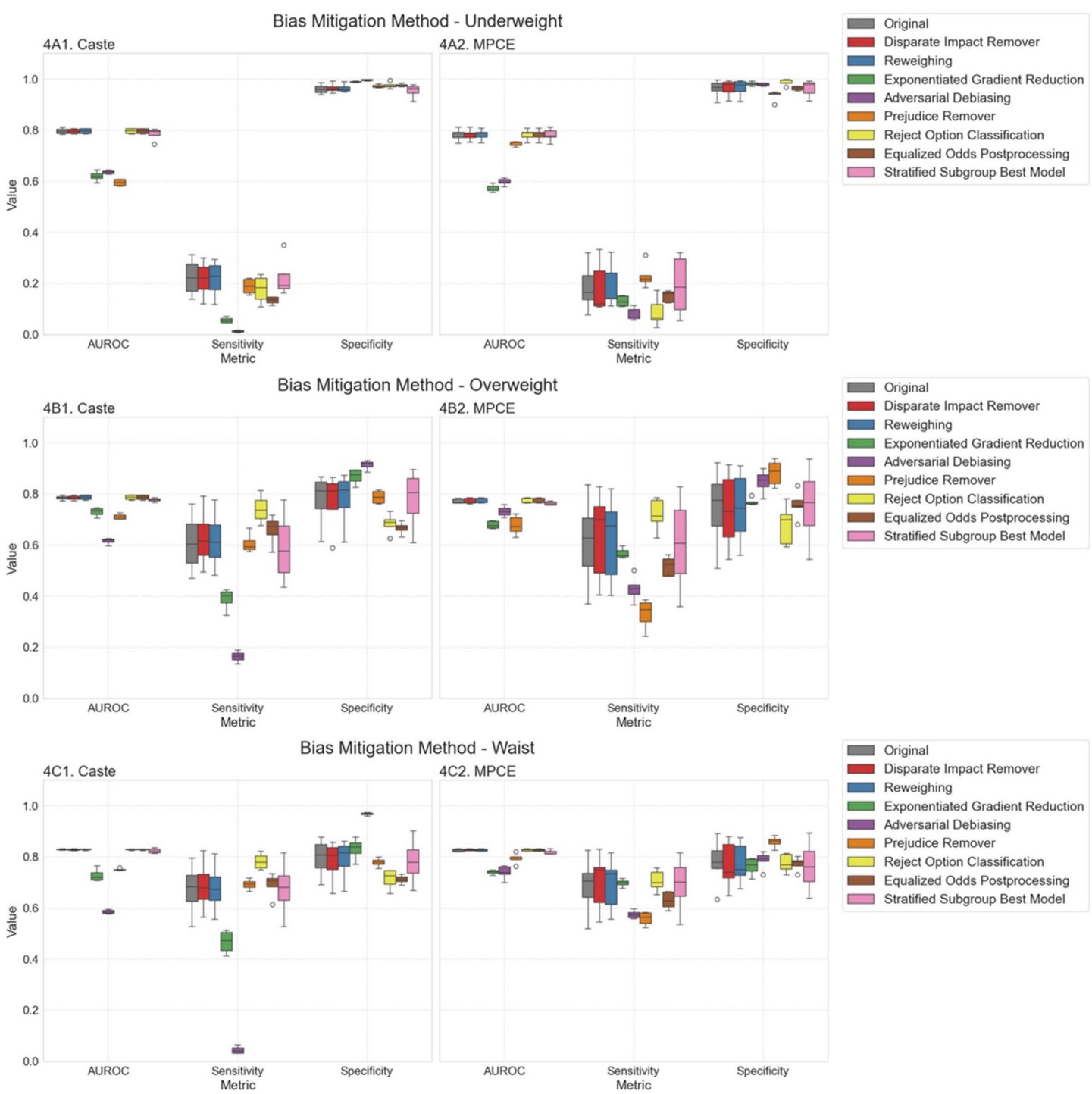

**Fig 4. Comparative Impact of Bias Mitigation Techniques on Subgroup Performance for Underweight, Overweight, and High Waist Circumference.**

subgroup variability—especially in sensitivity—their median AUROC and sensitivity values drop substantially, indicating a more pronounced degradation in overall performance. Meanwhile, Disparate Impact Remover Reweighting, and the Stratified Subgroup Best Model produce relatively modest changes, with box plots that remain close to the original grey distribution. Overall, Fig 4 underscores that although some methods can significantly narrow subgroup gaps—particularly for sensitivity and specificity—others introduce marked trade-offs in overall AUROC, demonstrating that different bias mitigation techniques vary widely in both their fairness benefits and their impact on predictive accuracy.

## Discussion

This study is the first to evaluate the machine learning techniques for predicting underweight, overweight, obesity, and central adiposity within the Indian context, while also addressing biases that related to socioeconomic and demographic disparities. Our findings demonstrate that tree-based algorithms, particularly LightGBM and Gradient Boosting, outperform others in predicting anthropometric measures, including underweight, overweight, and waist measurements. Incorporating socioeconomic and health-related significantly enhanced model performance, highlighting their importance in improving model accuracy. Key predictors identified, such as grip strength, gender, residence, and hypertension, emphasize the role of socioeconomic and health variables in shaping anthropometric outcomes.

Despite overall strong predictive capability, performance varied across demographic groups, with marginalized populations, such as lower socioeconomic groups and scheduled tribes, experiencing lower accuracy. These findings underscore the need to address biases in ML models to ensure fair and equitable health predictions across diverse demographic groups.

Bias mitigation strategies, including Reject Option Remover and Equalized Odds Postprocessing, improved fairness, whereas Exponentiated Gradient Reduction and Adversarial Debiasing reduced disparities but at the cost of overall accuracy. Methods like Disparate Impact Remover, Reweighting, and the Stratified Subgroup Best Model yielded minimal improvements.

### Comparison with existing literature

Our study advances research by using machine learning to predict nutritional outcomes—underweight, overweight, obesity, and adiposity—in a large, nationally representative sample of older Indian adults. While previous studies focused on obesity prediction in specialized datasets, such as using deep learning on environmental factors [25] or combining neuroimaging with metabolomics [26], our work uniquely addresses nutritional risk in later life, where both under- and overnutrition coexist. Additionally, by incorporating culturally specific factors like socioeconomic status and caste, we provide a level of contextual nuance often overlooked in prior studie[s 27–29].

Our approach is further distinguished by its rigorous evaluation and mitigation of algorithmic bias. Whereas prior studies using adversarial training have focused on reducing bias in models predicting outcomes like foregone care or cost-based risk among clinical populations [30] and analyses of widely deployed risk prediction tools have revealed significant racial biases arising from label choice [20] our work systematically implements a suite of debiasing techniques—including Disparate Impact Remover, Exponentiated Gradient Reduction, Re-sampling/Re-weighting, and Adversarial Training— complemented by stratified modelling. These measures help ensure that predictive accuracy is maintained equitably across marginalized subgroups (e.g., females, scheduled tribes, lower socioeconomic strata). In addition, although obesity risk prediction models developed in Bangladesh using relatively small datasets have shown high accuracy via logistic regression [31] our study not only achieves competitive performance but does so while addressing fairness in a heterogeneous, underexplored older population. Collectively, our integrated framework of high predictive performance and robust fairness adjustment fills an important gap in the literature, offering actionable insights for equitable public health interventions in low- and middle-income settings.

## Policy implications

Our findings highlight the potential of ML models to inform national health policies by predicting underweight, obesity, and adiposity with high accuracy. Algorithms like LightGBM show promise in predicting health outcomes such as underweight, obesity, and adiposity by incorporating socioeconomic and demographic variables. Key predictive features—grip strength, gender, residence, education, and chronic conditions—enable targeted, population-specific interventions. However, ML models often struggle struggle with performance disparities in marginalized groups, necessitating fairness-enhancing techniques to ensure equitable predictions. Addressing these gaps requires fairness-enhancing techniques, such as the Disparate Impact Remover, to ensure consistent model accuracy across diverse groups [19,21].

This study has three main limitations. First, Our study did not use survey weights, as it is still unclear how to properly include them in machine learning studies [32]. Second, the use of self-reported data introduces potential biases, such as recall bias or social desirability bias, which could impact the accuracy of key variables. Third, because LASI samples adults aged 45 years and older—and in our analytic cohort the vast majority are 45–75 years (≈90%; S1 Table)—we did not analyze model performance by age subgroup, and generalizability to younger populations is limited. Future research should explore improved weighting methods and validate findings using additional datasets for broader generalizability.

## Conclusion

ML models offer significant promise in predicting obesity and related conditions in India, but addressing biases in predictions is crucial for equitable health outcomes. Our study underscores the need for fairness-driven ML approaches in public health, particularly in the context of India's diverse population, providing actionable insights for more inclusive and effective policy decisions.

## Materials and methods

### Ethics statement

The study was approved by the NTU Ethical Review Board (NTU REC-No.: 202407HS010). Ethical approval for LASI data collection was granted by the Indian Council of Medical Research and the Institutional Review Board (IRB) at IIPS, Mumbai. Informed consent was obtained at household and individual levels.

### Data

This study uses a cross-sectional analysis of the Longitudinal Ageing Study in India (LASI) for the wave 2017–2018. LASI is a nationally representative survey of individuals aged 45 and above from all Indian states and union territories. The survey examines the health, social, and economic well-being of older adults. Harmonized with datasets like NHANES, ELSA, and CHARLS, LASI facilitates cross-population comparisons. Initially, the survey included 72,250 older adults and their spouses, enrolled through multistage probability sampling. After excluding 6,350 cases due to proxy interviews and participants younger than 45, the dataset was reduced to 64,867 individuals. Further data cleaning and processing to address incomplete records resulted in a final analytical sample of 55,647 participants.

The dataset provides detailed demographic, socioeconomic, and health-related information, including anthropometric measures such as Body Mass Index (BMI) and waist circumference (WC), collected through direct physical examinations. These data are critical for developing and validating machine learning models to identify obesity and related health risks. Data collection involved computer-assisted personal interviews, molecular biomarkers (e.g., dried blood spots), and non-molecular measures at the individual, household, and community levels. Variables used in this study are detailed in S2 Table.

## Measurements

The primary outcome measures include BMI and WC categories based on Asian standards for the Indian population: (1) BMI categories: underweight (<18.5 kg/m$^2$), normal weight (18.5–22.9 kg/m$^2$), and overweight/obesity (≥23 kg/m$^2$). (2) WC cut-offs: Men: >90 cm; Women: >80 cm.

## Pre-processing of independent variables

The independent variables underwent systematic pre-processing to improve data quality and facilitate model interpretability. All pre-processing procedures, including handling missing data, variable encoding, standardization, and feature engineering, were documented and are accessible on GitHub (https://github.com/johntayuleeHEPI/LASI-BMI.git). More detail descriptions of all the variables can be found in S2 Table. After feature derivation, categorical variables were encoded using one-hot encoding, and rows containing missing values were excluded from the dataset.

## Machine learning algorithms

We employed multiple supervised machine learning (ML) models, including tree-ensemble models like Random Forest, eXtreme Gradient Boosting (XGBoost), Gradient Boosting, Light Gradient Boosting Machine (LightGBM), and neural network models such as Deep Neural Networks (DNN), and Fully Convolutional Networks (FCN). Tree-ensemble models are broadly implemented for EHR-based risk stratification and adverse-event prediction [33], and for chronic-disease risk modelling [34], while deep neural networks are deployed in imaging workflows such as diabetic-retinopathy screening [35] and chest-X-ray CAD for tuberculosis [36]. The dataset was divided into training (80%) and testing (20%) subsets, with the models trained on the training set and evaluated on the testing set. Features were standardized by calculating the z-score for each data point within the respective feature.

## Model evaluation

The evaluation of model performance was conducted using the receiver operating characteristic (ROC) curve, a graphical representation of the trade-off between true positive rates (TPR) and false positive rates (FPR) across varying classification thresholds. The area under the ROC curve (AUROC) served as the primary metric to quantify the models' discriminative ability, indicating their effectiveness in distinguishing between classes. The model with the highest AUROC was selected for further in-depth analysis.

To ensure a comprehensive evaluation, additional performance metrics, including accuracy, sensitivity, specificity, and precision, were computed. Notably, sensitivity and specificity were emphasized for their importance in identifying true positive and true negative cases relative to the ground truth. These metrics were further stratified and analyzed across subgroups to assess variations in model performance under different conditions.

All performance and fairness metrics were estimated using 1,000 bootstrap iterations to derive confidence intervals (CIs), ensuring robust and reliable estimates of the models' capabilities and fairness across subgroups.

## Prediction density analysis

To evaluate the models' discriminative power, prediction density analysis was conducted using density plots for positive and negative classes. These plots provide insights into the model's sensitivity and specificity across subgroups. A left-skewed distribution for the positive class indicates high sensitivity, reflecting strong confidence in positive predictions, while a right-skewed distribution for the negative class signifies high specificity. In contrast, overlapping or flat distributions indicate poor class separation and limited predictive performance.

## Feature importance analysis

Feature importance was analysed using SHAP (SHapley Additive exPlanations) values. We use SHAP as our primary interpretability tool because it yields additive, consistent, local attributions grounded in Shapley values, suitable for

complex, non-linear estimators; while Permutation importance remains complementary but can produce biased or attenuated importances when features are correlated, a well-documented caveat in methodological guidance [37]. SHAP values, grounded in game theory, provide a detailed attribution of each feature's contribution to individual predictions. These methods enhanced model interpretability by identifying key factors influencing predictions.

Additionally, features were categorized into three groups: demographic, socioeconomic, and health-related variables. Three models were constructed, each incorporating a different set of features: demographic-only, demographic + socioeconomic, and demographic + socioeconomic + health. This approach enabled the examination of how the inclusion of different feature categories impacted model performance and interpretability.

## Debias and fairness in socioeconomic and caste subgroup

We evaluated performance and fairness across caste (Scheduled Castes, Scheduled Tribes, Other Backward Classes, and General) and MPCE quintiles given their legal salience and consistent association with disparities in health outcomes and service use in India [1,38,39]. In addition, to address the biases identified during the validation phase, we modified the training data to ensure balanced representation across all groups. The bias mitigation techniques were applied in three distinct stages: pre-processing, in-processing, and post-processing.

In the pre-processing stage, we employed Disparate Impact Remover [40], and Reweighting [41] methods to adjust the dataset. In the in-processing stage, Exponential Gradient Reduction [42] was applied to reduce bias during model training. For post-processing, we implemented fairness adjustments using Calibrated Equalized Odds Postprocessing [43], Reject Option Classification, and Equalized Odds Postprocessing. These methods were implemented using IBM's AI Fairness 360 (AIF360) [44], owing to its open-source, transparent design, wide adoption, and ease of implementation [45].

Additionally, we incorporated a stratified machine learning model to address potential arithmetic bias. The performance of this model was compared across various fairness adjustments to evaluate their overall impact on predictive accuracy. Notably, most bias mitigation approaches in our study relied on LightGBM following data pre-processing, with the exceptions of the Prejudice Remover—which defaults to logistic regression—and Adversarial Debiasing, which employs a deep learning framework. This integrated analysis enabled us to systematically assess the trade-offs between fairness and overall model performance.

## Patient and public involvement

Patients and/or the public were not involved in the design, conduct, reporting, or dissemination plans of this research.

## Supporting information

**S1 Table. Descriptive Statistics of Key Health Outcomes: Prevalence of Underweight, Overweight, and High Waist Circumference Based on BMI and Waist Measurements.**
(DOCX)

**S2 Table. Definitions and Descriptions of Variables Used in the Analysis.**
(DOCX)

**S3 Table. Comparison of Evaluation Metrics Across Different Machine Learning Models. (a) Underweight, (b) Overweight/Obesity, (c) High Waist Circumference.**
(DOCX)

**S4 Table. Performance of the Subgroup-Stratified models. (a) Underweight, (b) Overweight/Obesity, (c) High Waist Circumference.**
(DOCX)

**S5 Table. Comparison of Fairness Metrics (Equalized odds) Among Different Bias Mitigation Methods.**
(DOCX)

## Author contributions

**Conceptualization:** John Tayu Lee.

**Data curation:** Sheng Hui Hsu, Kanya Anindya.

**Formal analysis:** Sheng Hui Hsu, Meng-Huan Chen.

**Methodology:** Sheng Hui Hsu, Vincent Cheng-Sheng Li, Meng-Huan Chen.

**Supervision:** John Tayu Lee.

**Writing – original draft:** John Tayu Lee, Vincent Cheng-Sheng Li.

**Writing – review & editing:** Charlotte Wang, Toby Kai-Bo Shen, Valerie Tzu Ning Liu, Hsiao-Hui Chen, Rifat Atun.

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
