## [Decision Letter · Decision Letter 0]

15 Aug 2025

Response to Reviewers
Revised Manuscript with Track Changes
Manuscript
**Journal Requirements:**

i. State the initials, alongside each funding source, of each author to receive each grant.

ii. State what role the funders took in the study. If the funders had no role in your study, please state: “The funders had no role in study design, data collection and analysis, decision to publish, or preparation of the manuscript.”

2. Please ensure that your Ethics Statement is available in its entirety at the beginning of your Methods section, under a subheading 'Ethics Statement'.

3. Please upload separate figure files in .tif or .eps format. Also, remove the figures from your manuscript file but keep the legends.

4. We notice that your supplementary figures and tables are included in the manuscript file. Please remove them and upload them with the file type 'Supporting Information'. Please ensure that each Supporting Information file has a legend listed in the manuscript after the references list.

5. We have noticed that you have uploaded Supporting Information files, but you have not included a list of legends. Please add a full list of legends for your Supporting Information files after the references list.

**Additional Editor Comments (if provided):**
**Reviewers' Comments:**

**Comments to the Author**

1. Does this manuscript meet PLOS Digital Health’s publication criteria?

Reviewer #1: Yes

Reviewer #2: Yes

2. Has the statistical analysis been performed appropriately and rigorously?

Reviewer #1: I don't know

Reviewer #2: Yes

3. Have the authors made all data underlying the findings in their manuscript fully available (please refer to the Data Availability Statement at the start of the manuscript PDF file)?

Reviewer #1: Yes

Reviewer #2: Yes

4. Is the manuscript presented in an intelligible fashion and written in standard English?

Reviewer #1: Yes

Reviewer #2: Yes

Reviewer #1: This is very timely research and advances the field by identifying the gaps in ML technique application in a healthcare context. My expertise is in the health domain, and I have reviewed the manuscript from a healthcare application perspective. I would be relying on ML expert reviewers on the robustness of the ML application. From a healthcare application perspective, there are two key components that I would like to see in the paper to make it more relevant and informative for healthcare audiences:

Clarify criteria for subgroup selection: Please elaborate on the rationale for identifying and selecting certain groups as marginalized sub-populations. Explicitly stating the basis for these choices will help readers understand the context and strengthen the justification for your approach.

Provide real-world implementation examples: Could you expand on where these machine learning techniques have been applied in real-world settings? Including concrete examples, particularly in healthcare contexts, would help readers better appreciate the relevance and potential impact of your work.

Reviewer #2: This paper addresses a timely and important issue, in the context of public health and nutrition transition in India, as well as equity in AI in healthcare. The study is strengthened by the large, representative dataset and the range of algorithms it utilized for analysis. Finally, its use of socioeconomic and demographic variables provide a nuanced context for the population being studied. There are a few key points that should be addressed and refined by the authors.

Can the authors define the acronym “NML” in the introduction?

Only adults aged 45 and above are included in the study. In the introduction, can the authors give more information on the dual public health problem of underweight/overweight in India? What are characteristics of this issue in the age group being studied compared to younger population subgroups? Why is this a priority age demographic?

Can the authors describe why they used SHAP values versus permutation importance when analyzing feature importance?

Did the authors analyze model performance by age subgroup? With an age range as substantial as this study’s (45-116 years), I imagine there are distinct physiological differences between adults at various life stages. The paper would at minimum be strengthened by a discussion of this issue.

Can the authors include a brief 1-2 paragraph summary of their study population’s demographic characteristics at the start of the results section? It would be useful for framing the findings from the machine learning analysis.

Can the authors describe their rationale for choosing the selected bias mitigation methods?

The authors do not discuss the age of the participants as a limitation to the study’s generalizability. This is challenging as India has a rapidly growing population that is demographically young. The author’s thoughts on this study’s relevance to younger populations and the limits of this study should be discussed at a minimum.

**Do you want your identity to be public for this peer review?** For information about this choice, including consent withdrawal, please see our Privacy Policy

Reviewer #1: No

Reviewer #2: No

**Figure resubmission:****Reproducibility:** To enhance the reproducibility of your results, we recommend that authors of applicable studies deposit laboratory protocols in protocols.io, where a protocol can be assigned its own identifier (DOI) such that it can be cited independently in the future. Additionally, PLOS ONE offers an option to publish peer-reviewed clinical study protocols. Read more information on sharing protocols at https://plos.org/protocols?utm_medium=editorial-email&utm_source=authorletters&utm_campaign=protocols

---

## [Decision Letter · Decision Letter 1]

28 Sep 2025

Evaluating Fairness of Machine Learning Models in Predicting Underweight, Overweight, and Adiposity Across Socioeconomic and Caste Groups in India: Evidence from the Longitudinal Ageing Study in India

PDIG-D-25-00431R1

Dear Dr. Lee,

We're pleased to inform you that your manuscript has been judged scientifically suitable for publication and will be formally accepted for publication once it meets all outstanding technical requirements.

Within one week, you'll receive an e-mail detailing the required amendments. When these have been addressed, you'll receive a formal acceptance letter and your manuscript will be scheduled for publication.

An invoice for payment will follow shortly after the formal acceptance. To ensure an efficient process, please log into Editorial Manager at https://www.editorialmanager.com/pdig/ click the 'Update My Information' link at the top of the page, and double check that your user information is up-to-date. For billing related questions, please contact billing support at https://plos.my.site.com/s/.

Kind regards,

Po-Chih Kuo, Ph. D.

Section Editor

PLOS Digital Health

Additional Editor Comments (optional):

Reviewers' comments:

Reviewer's Responses to Questions

**Comments to the Author**

Reviewer #1: All comments have been addressed

Reviewer #2: All comments have been addressed

publication criteria?

Reviewer #1: Yes

Reviewer #2: Yes

3. Has the statistical analysis been performed appropriately and rigorously?

Reviewer #1: I don't know

Reviewer #2: Yes

4. Have the authors made all data underlying the findings in their manuscript fully available (please refer to the Data Availability Statement at the start of the manuscript PDF file)?

Reviewer #1: Yes

Reviewer #2: Yes

5. Is the manuscript presented in an intelligible fashion and written in standard English?

PLOS Digital Health does not copyedit accepted manuscripts, so the language in submitted articles must be clear, correct, and unambiguous. Any typographical or grammatical errors should be corrected at revision, so please note any specific errors here.

Reviewer #1: Yes

Reviewer #2: Yes

Reviewer #1: (No Response)

Reviewer #2: Reviewer comments have all been addressed

**Do you want your identity to be public for this peer review?** For information about this choice, including consent withdrawal, please see our Privacy Policy

Reviewer #1: No

Reviewer #2: No
